# Uncovering the Effects of Ammonium Sulfate on Neomycin B Biosynthesis in *Streptomyces fradiae* SF-2

**Xiangfei Li** [1,†]**, Fei Yu** [2,†]**, Kun Liu** [1]**, Min Zhang** [1]**, Yihan Cheng** [1]**, Fang Wang** [1]**, Shan Wang** [1]**, Rumeng Han** [1] **and Zhenglian Xue** [1,*]

1   Engineering Laboratory for Industrial Microbiology Molecular Beeding of Anhui Province, College of Biologic & Food Engineering, Anhui Polytechnic University, Wuhu 241000, China
2   Key Laboratory of Industrial Biotechnology, Ministry of Education, School of Biotechnology, Jiangnan University, Wuxi 214122, China
\*   Correspondence: xzlahpu@163.com
†   These authors contributed equally to this work.

**Abstract:** The aminoglycoside antibiotic neomycin has broad antibacterial properties and is widely used in medicine and agriculture. With the discovery of neomycin's potential applications in treating tumors and SARS-CoV-2, it is necessary to accelerate the biosynthesis of neomycin. In the present study, we investigated the effects of various inorganic salts on neomycin B (the main active neomycin) biosynthesis in *Streptomyces fradiae* SF-2. We found that 60 mM $(NH_4)_2SO_4$ could promote neomycin B biosynthesis and cell growth most effectively. Further comparative transcriptomic analyses revealed that 60 mM $(NH_4)_2SO_4$ inhibited the EMP and TCA cycles and enhanced the expression of *neo* genes involved in the neomycin B biosynthesis pathway. Finally, a neomycin B potency of 17,399 U/mL in shaking flasks was achieved by overexpressing *neoE* and adding 60 mM $(NH_4)_2SO_4$, corresponding to a 51.2% increase compared with the control *S. fradiae* SF-2. In the present study, the mechanism by which $(NH_4)_2SO_4$ affects neomycin biosynthesis was revealed through transcriptomics, providing a reference for the further metabolic engineering of *S. fradiae* SF-2 for neomycin B production.

**Keywords:** *S. fradiae*; neomycin B; $(NH_4)_2SO_4$; *neoE*; transcriptomic analysis

## 1. Introduction

    *Streptomyces fradiae* is a species of *Actinomycetota Phylum*, a genus of filamentous bacteria of the family *Streptomycetaceae*. The strain is a Gram-positive bacterium with a genomic G+C content of up to 70–74% [1–3]. The precursor compounds and energy generated by primary metabolism can be used to synthesize a variety of secondary metabolites with complex structures, diverse functions, and various biological activities [4]. Neomycin, a secondary metabolite first isolated from *S. fradiae* (GenBank: GCA_008704425.1) in the 1940s by Waksman, is a classical aminoglycoside antibiotic made from carbohydrates through the pentose phosphate pathway [5], which is bactericidal against both Gram-positive and Gram-negative bacteria, including *Staphylococcus aureus*, *Escherichia coli*, *Haemophilus influenzae*, *Proteus* spp., and *Serratia* spp. The antibiotic includes three classes, A, B, and C, each with its typical chemical structure and functions, including biological activities. Neomycin B, the main active neomycin, exhibits higher antimicrobial activity than other classical antibiotics (streptomycin, bacitracin, etc.). Neomycin B can bind to the 16S rRNA site of the 30S ribosome and interfere with the synthesis of bacterial proteins [5–7]. In recent years, with the development of medical diagnostic and clinical intervention technology, the potential function of neomycin B has become apparent in the medical field. To prevent respiratory and intestinal bacterial infections in livestock, neomycin can be added to the feed [8]. Additionally, neomycin can promote tumor cell apoptosis by activating the inhibitory factor p53 in tumor cells [9,10]. Furthermore, a recent study has shown that neomycin may inhibit SARS-CoV-2 as the protease inhibitor [11].

The biosynthesis of neomycin by *S. fradiae* is strongly affected by the composition of the medium. Recent studies have reported optimization of the fermentation medium to promote *Streptomyces* growth and neomycin biosynthesis [12,13]. In addition, some studies have shown that ammonium ions also affect growth and metabolism [14]. It was found that the nitrogen source affects growth and nitrate depletion causes biphasic growth patterns in batch cultures of strain OB3b [15]. To balance carbon and nitrogen metabolism, bacteria have evolved complex mechanisms to sense the nutrient supply and adapt their metabolism accordingly [16]. It has been demonstrated that nitrogen compounds, such as sodium nitrate, aspartic acid, and glutamic acid, could promote the growth of *S. fradiae* 3535 and neomycin production [17]. The results of these studies indicated that nitrogen metabolism and its metabolic intermediates might affect the central metabolism of methanotrophs, thereby controlling their growth. The biosynthesis of neomycin B in *S. fradiae* is mainly regulated by the *neo* gene clusters, which contain two operons, one comprised of 12 genes from *neoE* to *neoD* (responsible for neomycin B biosynthesis) and the other containing *neoGH–aphA* and other regulatory genes [1]. At present, metabolic engineering and other technical methods are being studied to improve neomycin production by enhancing its biosynthesis. For example, the overexpression of two regulatory genes of neomycin B biosynthesis (*afsA-g* and *neoR*) increased the neomycin B titer to $722.9 \pm 20.1$ mg/L and $564.7 \pm 32.5$ mg/L, respectively [18]. Moreover, an important conserved enzyme in 2-DOS synthesis is NeoE, a zinc-containing dehydrogenase that utilizes $NAD(P)^+$ as a coenzyme [19]. This enzyme belongs to the medium-chain dehydrogenase (MDR) family. MDR proteins contain a highly conserved structure of GxGxxG that could bind $NAD(P)^+$. Studies have shown that NeoE could catalyze the dehydrogenation of 2-deoxyscyllo inosamine (2-DOIA) to 3-amino-2,3-deoxyscyllo inosone (amino-DOI), and a *neoE* deletion mutant strain lost the ability to synthesize neomycin B [18]. Therefore, NeoE may be a key enzyme in the biosynthesis of neomycin B.

Comparative transcriptomics can be used to mine genes that are obviously positively regulated or negatively regulated by comparing and analyzing transcripts of samples in different states at different times [20,21]. Therefore, comparative transcriptomic sequencing is widely used to explore the mechanism of action on cellular metabolism under specific conditions [22]. For example, comparative transcriptomic analysis revealed that the transcription factor RosR could regulate L-glutamate metabolism and the L-glutamate biosynthesis network by interacting with promoter regions of many related genes [23]. Hence, we investigated the effects of different inorganic salts on neomycin B biosynthesis in *S. fradiae*. It was determined that 60 mM $(NH_4)_2SO_4$ most effectively promoted the biosynthesis of neomycin B and growth in *S. fradiae*. Next, we conducted a comparative transcriptomic analysis, which revealed that $(NH_4)_2SO_4$ affects neomycin B potency. Based on comparative transcriptomic analysis, the expression of related genes (*neo* gene cluster) involved in neomycin B was enhanced. In addition, the neomycin B potency of 17,399 U/mL at shake-flask was achieved by overexpressing the *neoE* gene and supplying 60 mM $(NH_4)_2SO_4$, which was 51.2% higher than that of the control *S. fradiae* SF-2. The present study reveals $(NH_4)_2SO_4$'s effect on neomycin B biosynthesis through transcriptomics, providing a good reference for the further metabolic engineering of *S. fradiae* SF-2 for neomycin B production.

## 2. Materials and Methods

### 2.1. Strains, Plasmids, and Growth Conditions

The strains and plasmids used in this work are listed in Table 1. The primers used in this study are listed in Table 2. With the primer pair NeoE F/NeoE R, the target fragment NeoE was amplified using the genome of *S. fradiae* SF-2 as a template. *S. fradiae* SF-2 was obtained through ARTP mutagenesis in the laboratory, and the whole genome sequencing was analyzed (data unpublished). To construct the PPR-NeoE recombinant plasmid, the vector pSET152 (PPR) was digested with *Not*I and *Eco*RV and then connected to the target DNA fragments by T4 DNA ligase (TaKaRa). Other recombinant plasmids were constructed

in the same way with the corresponding primers. All the recombinant plasmids constructed were transformed into *E. coli* DH5α and confirmed by colony PCR and sequencing.

**Table 1.** Strains and plasmids used in this study.

| Strains and Plasmids | Description | Source |
|---|---|---|
| Strains | | |
| *E. coli* DH5α | General cloning host | The lab |
| *E. coli* ET12567 | Demethylated strain containing pUZ8002 plasmid for conjugative transfer with actinomycetes | The lab |
| *S. fradiae* SF-2 | *Streptomyces fradiae*, neomycin B-producing strains generated by ARTP | [24] |
| ET12567/PPR-NeoB | *E. coli* ET12567 derivative harboring PPR-NeoB | This study |
| ET12567/PPR-NeoE | *E. coli* ET12567 derivative harboring PPR-NeoE | This study |
| ET12567/PPR-NeoP | *E. coli* ET12567 derivative harboring PPR-NeoP | This study |
| ET12567/PPR-NeoQ | *E. coli* ET12567 derivative harboring PPR-NeoQ | This study |
| ET12567/PPR-NeoS | *E. coli* ET12567 derivative harboring PPR-NeoS | This study |
| ET12567/PPR-NeoL | *E. coli* ET12567 derivative harboring PPR-NeoL | This study |
| ET12567/PPR-NeoC | *E. coli* ET12567 derivative harboring PPR-NeoC | This study |
| ET12567/PPR-NeoD | *E. coli* ET12567 derivative harboring PPR-NeoD | This study |
| ET12567/PPR-NeoF | *E. coli* ET12567 derivative harboring PPR-NeoF | This study |
| ET12567/PPR-NeoM | *E. coli* ET12567 derivative harboring PPR-NeoM | This study |
| ET12567/PPR-NeoN | *E. coli* ET12567 derivative harboring PPR-NeoN | This study |
| SF-NeoB | *S. fradiae* SF-2 derivative the expression of NeoB | This study |
| SF-NeoE | *S. fradiae* SF-2 derivative the expression of NeoE | This study |
| SF-NeoP | *S. fradiae* SF-2 derivative the expression of NeoP | This study |
| SF-NeoQ | *S. fradiae* SF-2 derivative the expression of NeoQ | This study |
| SF-NeoS | *S. Fradiae* SF-2 derivative the expression of NeoS | This study |
| SF-NeoL | *S. fradiae* SF-2 derivative the expression of NeoL | This study |
| SF-NeoC | *S. fradiae* SF-2 derivative the expression of NeoC | This study |
| SF-NeoD | *S. fradiae* SF-2 derivative the expression of NeoD | This study |
| SF-NeoF | *S. fradiae* SF-2 derivative the expression of NeoF | This study |
| SF-NeoM | *S. fradiae* SF-2 derivative the expression of NeoM | This study |
| SF-NeoN | *S. fradiae* SF-2 derivative the expression of NeoN | This study |
| Plasmids | | |
| PPR (pSET152-PermE*) | *E. coli-S. fradiae* shuttle vector for the expression of target protein | The lab |
| PPR-NeoB | Derived from PPR, for the expression of NeoB | This study |
| PPR-NeoE | Derived from PPR, for the expression of NeoE | This study |
| PPR-NeoP | Derived from PPR, for the expression of NeoP | This study |
| PPR-NeoQ | Derived from PPR, for the expression of NeoQ | This study |
| PPR-NeoS | Derived from PPR, for the expression of NeoS | This study |
| PPR-NeoL | Derived from PPR, for the expression of NeoL | This study |
| PPR-NeoC | Derived from PPR, for the expression of NeoC | This study |
| PPR-NeoD | Derived from PPR, for the expression of NeoD | This study |
| PPR-NeoF | Derived from PPR, for the expression of NeoF | This study |
| PPR-NeoM | Derived from PPR, for the expression of NeoM | This study |
| PPR-NeoN | Derived from PPR, for the expression of NeoN | This study |

*S. fradiae* SF-2, *E. coli* DH5α, and *E. coli* ET12567 were deposited in the laboratory. *E. coli* DH5α was used as a host cell for cloning. *E. coli* ET12567 was used for conjugative transfer with actinomycetes. *S. fradiae* SF-2 was grown with AS-1 solid medium at 35 °C [25]. *E. coli* DH5α and *E. coli* ET12567 were grown in LB medium at 37 °C. *S. fradiae* SF-2 single colonies were first cultivated in seed medium at 35 °C and 260 rpm to the log phase (40–50 h) and then seed spores were inoculated (1% *v/v*) into fermentation medium at 35 °C, 260 rpm, and 75% relative humidity for 7 days. AS-1 medium contained 1 g/L yeast powder, 0.2 g/L L-alanine, 0.2 g/L L-arginine, 0.5 g/L L-aspartate, 2.5 g/L NaCl, 10 g/L $Na_2SO_4$, 5 g/L soluble starch, and 20 g/L agar powder, pH 7.3–7.8. Seed medium contained 1 g/L $(NH_4)_2SO_4$, 20 g/L yeast powder, 10 g/L groundnut meal, 10 g/L soluble starch, 30 g/L glucose, 10 g/L corn steep liquor, 5 g/L trypsin, 1 g/L $Na_2HPO_4$, 10 g/L

CaCO$_3$, and 2 g/L bean oil, pH 7.3–7.8. The fermentation medium contained 70 g/L soluble starch, 28 g/L groundnut meal, 6 g/L yeast powder, 6 g/L (NH$_4$)$_2$SO$_4$, 20 g/L glucose, 2.5 g/L corn steep liquor, 9 g/L trypsin, 5 g/L medium-temperature bean cake powder, 4.5 g/L NaCl, 0.3 g/L high-temperature amylase, 0.4 g/L Na$_2$HPO$_4$, 4 g/L CaCO$_3$, and 3 g/L bean oil, pH 6.8–7.3.

**Table 2.** Primers used in this study.

| Primer Name | Sequences | Digest Sites |
|---|---|---|
| NeoB F | ATAGCGGCCGCGATGACGAAAAACTCTTCCCTGC | NotI |
| NeoB R | CGAGATATCTCAGTCGTCCAGCAGCCG | EcoRV |
| NeoE F | ATAGCGGCCGCGATGAAGGCTCTGGTGTTCGAGG | NotI |
| NeoE R | CGAGATATCTCAGGCCCGGAGGTTGAAGTA | EcoRV |
| NeoP F | ATAGCGGCCGCGATGACGGCCGCCCAGC | NotI |
| NeoP R | CGAGATATCTCATGCCGTCCTGGCCAG | EcoRV |
| NeoQ F | ATAGCGGCCGCGATGAAGCGCCTTCGAGGCAC | NotI |
| NeoQ R | CGAGATATCTCAGACGTGCGCGGTGTGC | EcoRV |
| NeoS F | ATAGCGGCCGCGATGGTCTCCCCGTTGGCA | NotI |
| NeoS R | CGAGATATCTCAAGTGGCCAGGTCGGC | EcoRV |
| NeoL F | ATAGCGGCCGCGGTGGTGACGACCGGCGTGGC | NotI |
| NeoL R | CGAGATATCTCAGGCCAGTGCGGCGAC | EcoRV |
| NeoC F | ATAGCGGCCGCGATGCAGACCACCCGCAT | NotI |
| NeoC R | CGAGATATCTTACGGCACGGGTCCGGC | EcoRV |
| NeoD F | ATAGCGGCCGCGGTGGGTGAGCCGACGTGG | NotI |
| NeoD R | CGAGATATCTCACCGGGCACCCGCCG | EcoRV |
| NeoF F | ATAGCGGCCGCGGTGGCTGAGGCGCCTGC | NotI |
| NeoF R | CGAGATATCTCACCCACCGTGCTCCTCC | EcoRV |
| NeoM F | ATAGCGGCCGCGGTGCTGCGGCTCACCC | NotI |
| NeoM R | CGAGATATCTCACGGCGCCCACCCG | EcoRV |
| NeoN F | ATAGCGGCCGCGATGACCACCGAC | NotI |
| NeoN R | CGAGATATCTCATACGAGCG | EcoRV |
| RT-NeoE F | TGACGGCCACCTTCTCGC | |
| RT-NeoE R | ACCTGCTCCTGCGGCACCT | |
| RT-NeoS F | TAGGTGTAGTACGTACGGG | |
| RT-NeoS R | ATGGGCAGCAACCGCTGCCT | |
| RT-NeoC F | GTTCTTGACGAGCGCGGT | |
| RT-NeoC R | GGACACGCCATCGAGCACG | |
| RT-NeoD F | CTCGGTCTCGTCGTCGTA | |
| RT-NeoD R | ACGCGACGCTCCTGACGGTGT | |
| RT-NeoM F | TGGACGTGCACCAGGT | |
| RT-NeoM R | AGCAGCTCGTCATGACCGT | |
| RT-NeoN F | TGGTAGTTGTAGCCGTTGGT | |
| RT-NeoN R | ACTGCTCCACTTCATGCCGC | |

*2.2. Comparative Transcriptomic Sequencing*

*S. fradiae* SF-2 preserved at −80 °C was cultured on AS-1 solid medium for 5–7 days at 35 °C. Then, single colonies were inoculated in seed medium at 35 °C and 260 rpm to the log phase. Next, the cultured cells were transferred to fermentation medium with or without 60 mM (NH$_4$)$_2$SO$_4$ and cultured for 48 h at 35 °C and 260 rpm. Finally, the fermented media were centrifuged for 10 min at 4 °C and 8000 rpm. The cultured cells were collected, snap-frozen in liquid nitrogen, and sent to Shanghai Megji Biomedical Technology Co., Ltd. for comparative transcriptomic sequencing. For annotation, *S. fradiae* DSM 40063 (GenBank: AJ629247.1) was used as the reference.

*2.3. RT-qPCR Analysis*

RT-qPCR reactions were conducted with ChamQ Universal SYBR qPCR Master Mix*Q711-02 (Vazyme Biotech Co., Ltd., Nanjing, China) to confirm the validity of the RNA-seq data. The StepOnePlus 96 real-time PCR system (Applied Biosystems Inc., Waltham,

MA, USA) was used to amplify and quantify the PCR products. The program was as follows: 30 s at 95 °C, followed by 40 cycles of 10 s at 95 °C and 30 s at 60 °C. Relative transcript levels were calculated by the $2^{-\Delta\Delta Ct}$ method. RT-qPCR was tested with three reactions in parallel. The primers used for RT-qPCR analysis are listed in Table 2.

### 2.4. Optimizing the Conjugation between E. coli and S. fradiae SF-2

The spore suspension of *S. fradiae* SF-2 was incubated for 10 min at 50 °C and then for 3 h at 37 °C at 200 rpm. Next, the donor cells were mixed with a cultured spore suspension of *S. fradiae* SF-2 (donor cells: receptor cells = 10:1). Afterwards, the mixture was centrifuged at 4000 rpm for 5 min at 4 °C, spread on AS-1 solid medium with 75 mM $MgCl_2$, and incubated for 14 h at 30 °C. Next, 50 µg/mL apramycin and 500 µg/mL nalidixic acid were used to cover the AS-1 plate, followed by incubation at for 4–5 days at 30 °C.

### 2.5. Detection of Neomycin and Residual Sugar

A spectrophotometer (UV-1800) was used to determine the optical density at 600 nm ($OD_{600}$). Neomycin B potency was determined as described previously [1]. The method for detection of neomycin B (HPLC) was as follows: chromatographic Agilent C18 column, flow rate: 1 mL/min, flow phase: acetonitrile/water (95:5, *v*/*v*); temperature: 25 °C; injection volume: 10 µL; absorption: 265 nm. Reducing sugar levels were determined as previously described [23]. All assays were performed in triplicate.

## 3. Results and Discussion

### 3.1. $(NH_4)_2SO_4$ Promoted the Biosynthesis of Neomycin B

To investigate the effects of various inorganic salts on neomycin B biosynthesis, different concentrations (CK: without $(NH_4)_2SO_4$ addition, 20, 40, 60, and 80 mM) of NaCl, KCl, $(NH_4)_2SO_4$, and $K_2SO_4$ were added to the fermentation medium. The addition of inorganic salts affected the potency of neomycin B in shake flasks differently (Figure 1). In the presence of 60 mM $(NH_4)_2SO_4$, 80 mM NaCl, 40 mM KCl, and 60 mM $K_2SO_4$, the highest potency of neomycin B was achieved, corresponding to 13,650.0, 7429.7, 6574.1, and 6317.2 U/mL, respectively, which was 3.3, 0.82, 0.61, and 0.54 times higher than the control without inorganic salts. According to the results, it was found that the accumulation of neomycin B was promoted by adding 60 mM $(NH_4)_2SO_4$ more efficiently than in the other test groups, which might be attributed to three aspects. On the other hand, to control the fed-batch cultures, the culture phase has to be divided into three sections with different C/N ratios: initial, exponential, and neomycin production. First, $(NH_4)_2SO_4$ could increase the cell's osmotic pressure to efficiently utilize carbon and nitrogen sources, thereby enhancing neomycin biosynthesis. Second, the addition of $(NH_4)_2SO_4$ can moderately reduce the C/N ratios in the fermentation medium, which is quite beneficial for the growth and metabolism of the strain and further neomycin B production. Third, $(NH_4)_2SO_4$ may also act as an amino donor to increase the transaminase activity involved in neomycin B biosynthesis.

### 3.2. $(NH_4)_2SO_4$ Enhanced Cell Growth and Utilization of Reducing Sugar

In antibiotic production, nitrogen is an essential nutrient used by bacteria for cell growth and secondary metabolite synthesis. The bacteria can directly absorb and utilize appropriate amounts of inorganic nitrogen or organic nitrogen in the form of protein degradation products. The inorganic nitrogen $(NH_4)_2SO_4$ can facilitate the conversion of α-ketoglutarate to L-glutamate by the TCA cycle; L-glutamate is transformed into L-glutamine by transamination, thereby promoting cell growth [16,26–28]. To further investigate the effects of $(NH_4)_2SO_4$ on cell growth and neomycin B biosynthesis, the specific growth rate, the efficiency of neomycin biosynthesis, and the utilization of reducing sugar of *S. fradiae* SF-2 were determined in the presence or absence of 60 mM $(NH_4)_2SO_4$.

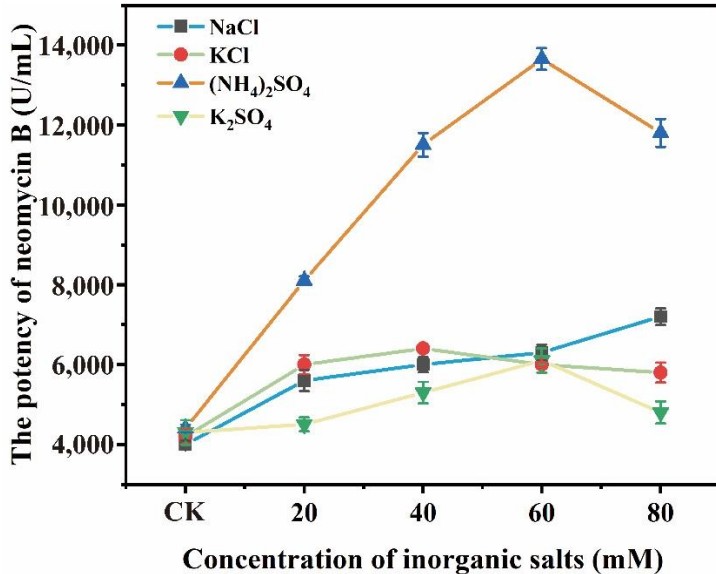

**Figure 1.** The effects of different inorganic salts on the biosynthesis of neomycin B.

It was found that 60 mM $(NH_4)_2SO_4$ promoted *S. fradiae* SF-2 growth, causing the cells to enter the logarithmic growth phase more quickly and reach a maximum specific growth rate of 0.122 $h^{-1}$ at about 24 h (Figure 2A). Moreover, *S. fradiae* SF-2 without $(NH_4)_2SO_4$ supplementation reached a maximum specific growth rate of 0.056 $h^{-1}$ at 36 h and entered the stable phase at 72 h. These results indicate that the addition of appropriate amounts of $(NH_4)_2SO_4$ could accelerate the growth of *S. fradiae* SF-2, which may be attributed to the fact that it reduced the C/N ratio in the fermentation medium. Secondly, in the presence of 60 mM $(NH_4)_2SO_4$, *S. fradiae* SF-2 began synthesizing neomycin B at 48 h, which significantly improved the efficiency of neomycin B biosynthesis compared to the control without $(NH_4)_2SO_4$. This may be related to the ability of $(NH_4)_2SO_4$ to alter the physical and chemical properties of the cell wall to promote the absorption of carbon and nitrogen sources by *S. fradiae* SF-2 used for the synthesis of secondary metabolites. This was consistent with previous studies that reported that the addition of ammonium boosts product biosynthesis, such as gentamicin in *Micromonospora purpurea*, glycopeptide A40926 in *Actinomadura* sp. ATCC 39727, and avilamycin in *Streptomyces viridochromogenes* [29–31]. Moreover, neomycin B biosynthesis requires several aminotransferases, and thus adding $(NH_4)_2SO_4$ may increase aminotransferase activity. However, excessive nitrogen sources can inhibit secondary metabolite biosynthesis. It was reported that when the $NH_4^+$ concentration exceeds 20 mM, valine dehydrogenase and glucose-6-phosphate dehydrogenase activity was inhibited, thereby decreasing erythromycin production [32]. Therefore, 60 mM $(NH_4)_2SO_4$ was found to be a suitable concentration for neomycin biosynthesis in this study. Third, at 60 mM $(NH_4)_2SO_4$, the final residual sugar content at the end of fermentation was 4.2 g/L, which was much lower than the condition without $(NH_4)_2SO_4$ (Figure 2B,C), suggesting that $(NH_4)_2SO_4$ could promote the utilization of reducing sugar in *S. fradiae* SF-2. In conclusion, $(NH_4)_2SO_4$ could improve neomycin B potency in *S. fradiae* by promoting the utilization of carbon sources.

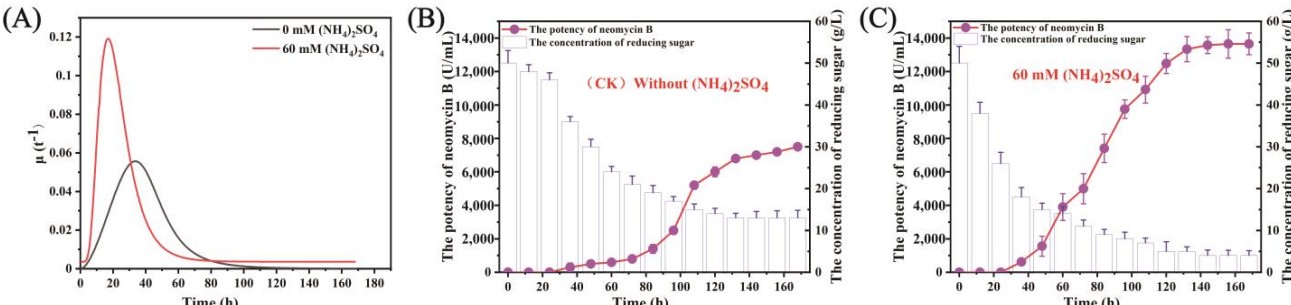

**Figure 2.** The effects of ammonium sulfate on the growth, reducing sugar utilization, and neomycin B biosynthesis in *S. fradiae* SF-2: (**A**) The effects of the addition of 60 mM $(NH_4)_2SO_4$ on the specific growth rate. (**B**,**C**) The impact of the addition of 60 mM $(NH_4)_2SO_4$ on reducing sugar utilization and neomycin B potency.

*3.3. Comparative Transcriptomics Revealed the Mechanisms Underlying the Effect of $(NH_4)_2SO_4$ on Neomycin B Biosynthesis*

To further investigate the mechanisms underlying the effect of $(NH_4)_2SO_4$ on neomycin B biosynthesis in *S. fradiae*, the cells were fermented for 48 h with and without 60 mM $(NH_4)_2SO_4$. The results showed that a total of 5902 genes were expressed in the medium with 60 mM $(NH_4)_2SO_4$ compared with the condition with no $(NH_4)_2SO_4$ addition, of which 637 were specifically expressed (Figure 3A), based on comparative transcriptomic analysis. Compared with the culture condition without $(NH_4)_2SO_4$, the expression levels of a total of 880 genes changed significantly, among which 651 genes were significantly upregulated and 229 genes were significantly downregulated (Figure 3B and Supplementary Materials). The results indicated that 60 mM $(NH_4)_2SO_4$ significantly influenced the gene expression of *S. fradiae* SF-2.

KEGG pathway enrichment analysis of the differentially expressed genes revealed that the genes related to metabolism were mainly involved in amino acid metabolism, carbohydrate metabolism, glycan biosynthesis and metabolism, cofactor and vitamin metabolism, nucleic acid metabolism, and energy metabolism. The genes differentially expressed related to the genetic information processing system are mainly involved in translation, replication, and repair. The differentially expressed genes related to environmental information processing are primarily involved in membrane transport and signal transduction (Figure 3C). The analysis also revealed that there was a significant difference in the expression of genes involved in the TCA cycle, oxidative phosphorylation, amino acid metabolism, propionate metabolism, carboxylic acid metabolism, polyketose metabolism, and vancomycin and staurosporine biosynthesis (Figure 3D). Functional enrichment analysis of the differentially expressed genes showed that they were largely involved in the biosynthesis and metabolism of xylulose-5-phosphate, monosaccharide decomposition, arabinose metabolism, pentose catabolism, and carbohydrate catabolism. The transcription levels of these genes were significantly changed, which may have significant effects on the biosynthesis of neomycin (Figure 3E). Finally, we found that the transcript levels of genes involved in the EMP and TCA cycles were significantly downregulated compared with the control without $(NH_4)_2SO_4$, indicating that the carbon metabolism flow was pushed toward neomycin biosynthesis (Figure 4). Moreover, the transcript levels of *neo* genes that regulate neomycin biosynthesis were upregulated compared to the control without $(NH_4)_2SO_4$. Furthermore, the upregulation of nitrogen assimilation-related genes also indicated that the nitrogen transport and utilization capacity were significantly improved by $(NH_4)_2SO_4$. Additionally, the genes related to pentose catabolism and carbohydrate catabolism were significantly upregulated, which provided precursors for the biosynthesis of neomycin B. To further verify the reliability of the comparative transcriptomic data, RT-qPCR analysis was performed on the *neo* gene cluster genes. The RT-qPCR results were consistent with the results of the comparative transcriptomic analysis (Figure 4), which indicated that the comparative transcriptomic analysis was reliable.

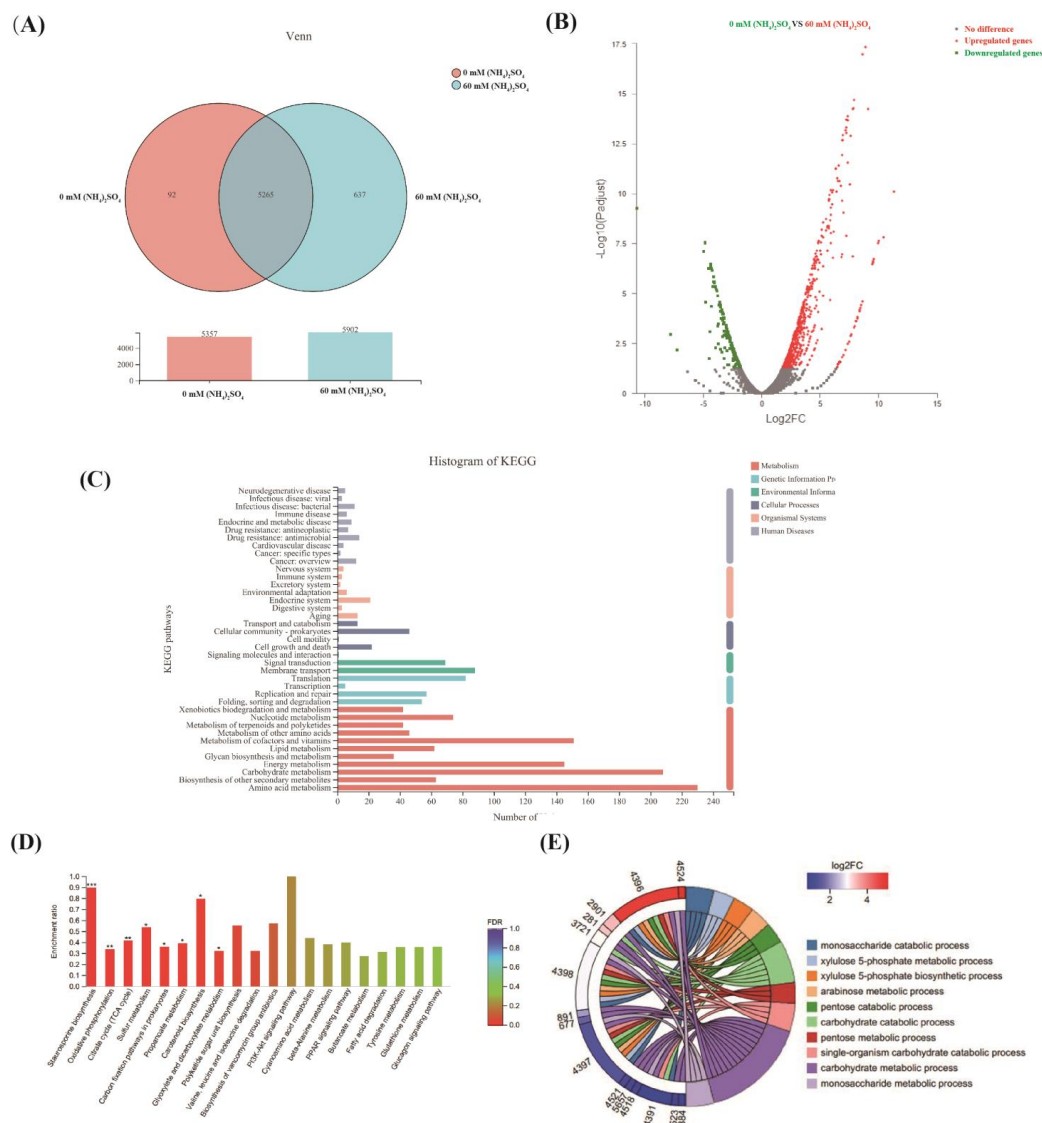

**Figure 3.** Comparative transcriptomic analysis of the effects of 60 mM $(NH_4)_2SO_4$ on neomycin B biosynthesis: (**A**) Venn diagram analysis of differentially expressed genes. (**B**) Volcano plot analysis of differentially expressed genes. The *x*-axis represents the $\log_2$-transformed expression fold-change values. The *y*-axis represents the $\log_{10}$-transformed adjusted *p*-values. Red dots indicate upregulated genes and blue dots indicate downregulated genes. (**C**) KEGG enrichment analysis of the metabolic pathways in which differentially expressed genes are involved. The *y*-axis is the name of the KEGG metabolic pathway, and the abscissa is the number of genes annotated to the pathway. KEGG metabolic pathways can be divided into seven major categories: metabolism, genetic information processing, environmental information processing, cellular processes, organismal systems, human diseases, and drug development. (**D**) KEGG enrichment analysis of differentially expressed genes. The X-axis is the name of the pathway. The Y-axis represents the enrichment rate, which is the ratio of the Sample number of genes annotated to the pathway and the background number of all genes annotated to the pathway. The higher the value of the rich factor, the greater the degree of enrichment. The color represents the enrichment significance (*p*-value), and the darker the color, the more significantly the pathway is enriched, where *p*-value < 0.001 is labeled as ***, *p*-value < 0.01 as **, *p*-value < 0.05 as *, and the color gradient on the right indicates the *p*-value size. (**E**) Functional enrichment analysis of differentially expressed genes. On the left is the gene, which is arranged in the order of $\log_2$FC from largest to smallest. Larger $\log_2$FC values indicate larger differential expression ploidy for upregulated genes. Smaller $\log_2$FC values indicate larger differential expression ploidy for downregulated genes. A $\log_2$FC closer to 0 indicates smaller differential expression ploidy for genes.

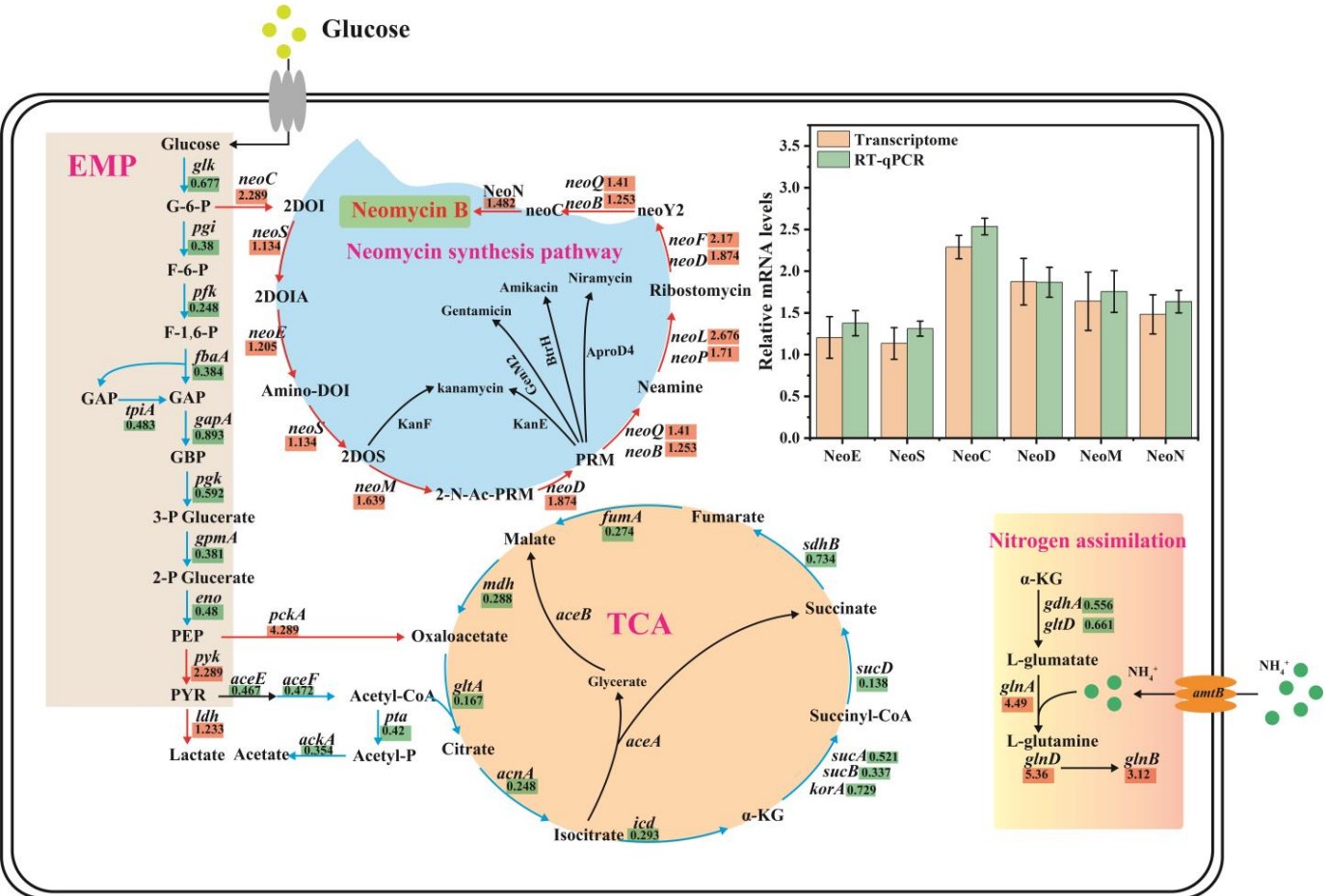

**Figure 4.** The expression levels of genes involved in neomycin B metabolism in *S. fradiae* SF-2 with 60 mM $(NH_4)_2SO_4$ relative to 0 mM $(NH_4)_2SO_4$. The red arrows show genes whose expression levels are upregulated, and the green arrows show those whose expression levels are downregulated.

### 3.4. neoE Overexpression Improved the Biosynthesis of Neomycin B

Using comparative transcriptomic analysis, it was found that $(NH_4)_2SO_4$ enhanced the expression of *neo* genes involved in neomycin B biosynthesis and reduced the expression of genes involved in the EMP and TCA cycles, thereby promoting neomycin B production. To further improve neomycin B potency, each gene in the *neo* gene clusters was overexpressed. Overexpression of *neoE*, *neoS*, *neoC*, *neoD*, *neoF*, *neoM*, and *neoN* significantly promoted the accumulation of neomycin B. Overexpression of *neoE* resulted in 15,810.8 U/mL of neomycin B after fermentation for 168 h, which was 37.5% higher than that achieved with the *S. fradiae* SF-2 control (Figure 5A). Finally, the engineered *S. fradiae* SF-*neoE* strain was fermented for 168 h with 60 mM $(NH_4)_2SO_4$ supplementation. A potency of 17,399 U/ mL was achieved, corresponding to a 51.2% increase compared to the wild-type strain *S. fradiae* SF-2 (Figure 5B).

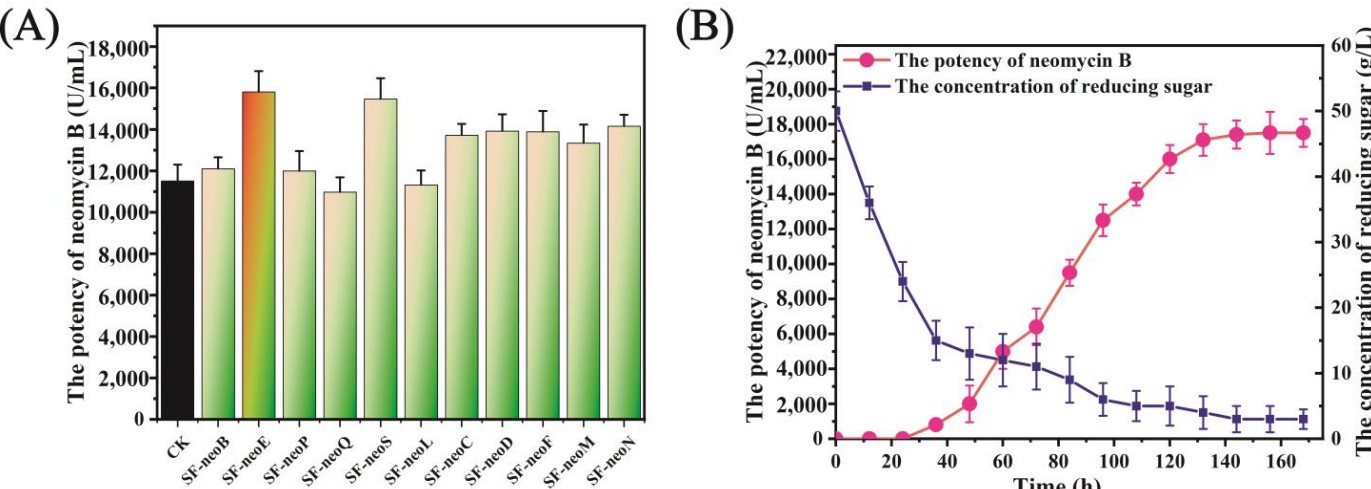

**Figure 5.** (**A**) The effects of overexpression of single genes of the *neo* gene clusters on neomycin B biosynthesis in *S. fradiae* SF-2. (**B**) The effect of 60 mM $(NH_4)_2SO_4$ on reducing sugar utilization and neomycin B biosynthesis in the SF-*neoE* strain.

## 4. Conclusions

This study proved that 60 mM $(NH_4)_2SO_4$ could inhibit the EMP and TCA cycles, promote the utilization of reducing sugars, and enhance the expression of *neo* genes involved in the neomycin B biosynthesis pathway, thereby improving the neomycin B potency. Upon *neoE* overexpression and the addition of 60 mM $(NH_4)_2SO_4$, the engineered *S. fradiae* SF-*neoE* strain presented a 51.2% increase (17,399 U/mL) compared with the control strain *S. fradiae* SF-2. In summary, uncovering the mechanisms underlying the effects of $(NH_4)_2SO_4$ on neomycin B biosynthesis in *S. fradiae* is beneficial for enhancing neomycin B production and applications.

**Supplementary Materials:** The following supporting information can be downloaded at: https://www.mdpi.com/article/10.3390/fermentation8120678/s1, Table S1: Raw data of comparative transcriptomic.

**Author Contributions:** Conceptualization, X.L., F.Y. and Z.X.; writing—original draft preparation, X.L. and F.Y.; methodology, K.L., M.Z., Y.C., F.W., S.W. and R.H.; supervision, Z.X. All authors have read and agreed to the published version of the manuscript.

**Funding:** This study was supported by the National Nature Science Foundation of China (31471615, 31871781, and 31772081).

**Institutional Review Board Statement:** Not applicable.

**Data Availability Statement:** The data that support the findings of this study are available from the corresponding author upon reasonable request.

**Conflicts of Interest:** The authors declare no conflict of interest.

## Abbreviations

EMP: glycolytic pathway; TCA: tricarboxylic acid cycle. RH: relative humidity.

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
