# Peer review of "Uncovering the Effects of Ammonium Sulfate on Neomycin B Biosynthesis in Streptomyces fradiae SF-2"

_fermentation, doi:10.3390/fermentation8120678_

Round 1

Reviewer 1 Report

Thank you for your submission.

Although the study might be interesting for the community, the work is not convincing. Please find below and attached my comments and concerns. 

Recommendation: publication in a journal with a lower IF.

Major concerns (Examples):

Line 2-3: Title

“Uncovering the mechanism of (NH4)2SO4 effect on neomycin B synthesis in Streptomyces fradiae SF-2”

The title is suboptimal.

The manuscript presents different effects of ammonium sulfate addition on the gene expression. It is not providing the reader with a clear explanation of the mechanism.  

The chemical formular in a title should be avoided (ammonium sulfate).

As the compound is not synthesized chemically, it should read “biosynthesis” not synthesis”. 

It is the same for abstract and the whole manuscript “biosynthesis” not “synthesis”. This should be adapted in the text.  

EMP and TCA: Abbreviations should be introduced

Line 22:

“…the neomycin B potency of 17399…”

What does it mean?

“level” should be deleted -> in shaking flasks

Line 24-25:

In this study, the mechanism of (NH4)2SO4 affects neomycin synthesis was revealed through transcriptomics, which pushed the development of neomycin B production and applications

Transcriptomics cannot accelerate the production and application of an antibiotic. Transcriptomics is a method…

The same sentence was repeated in the introduction.

Line 30-31:

The sentence “Streptomyces fradiae belongs to the bacterial domain, such as actinomycetes and streptomycetes” is not correct.

Domain: Bacteria

Actinomycetes: a group of Actinomycetales

Streptomyces: Genus

Actinomycetes and streptomycetes are not a domain.

Line 37-40:

“Especially for neomycin B, the main active ingredient in neomycin, it exhibits higher antimicrobial activity than other classical antibiotics (streptomycin and bacitracin, etc.) that could bind to the 16S rRNA site of the 30S ribosome and interfere with the synthesis of bacterial proteins [5-7].”

This sentence is confusing/incorrect. The main mode of action of bacitracin is the inhibits peptidoglycan synthesis (target phosphorylase). Therefore, the comparison is weird. I am not aware of the fact that bacitracin inhibits protein biosynthesis. Even if there are new data supporting it, the approval for medical use was based on the inhibition of peptidoglycan synthesis in bacteria.

In addition, the citations (see below) do not support the statement. At lease the titles and abstracts do not suggest that the publications describe the binding of bacitracin to “16S rRNA site of the 30S ribosome”.  

5. Hanko, V.P.; Rohrer, J.S. Suitability of a liquid chromatography assay of neomycin sulfate to replace the microbiological assay 301 for neomycin in USP monographs. J. Pharm. Biomed. Anal. 2010, 51, 96-102. 302

6. Waksman, S.A. ; Lechevalier, H.A. Neomycin, a new antibiotic active against streptomycin-resistant bacteria, including tuber- 303 culosis organisms. Science 1949, 109, 305-307. 304

7. Secombe, K.R.; Ball, I.A.; Wignall, A.D.; Bateman, E.; Keefe, D.M.; Bowen, J.M. Antibiotic treatment targeting gram negative 305 bacteria prevents neratinib-induced diarrhea in rats. Neoplasia 2022, 30, 100806.  

Line 37-40:

“Furthermore, a recent study has shown that neomycin B may inhibit COVID-19 [11].”

In this sentence the inhibition of SARS-CoV-2 is stated. However, in the citated publication neomycin was not mentioned at all (see publication)…

Rampogu, S.; Lee, K.W. Pharmacophore modelling-based drug repurposing approaches for SARS-CoV-2 therapeutics. Frontiers 313 in Chemistry 2021, 9.

Line 49-50:

“The composition of the medium has a significant impact on the synthesis of neomycin 47 by S. fradiae. Recent studies have reported optimizing the fermentation media to promote 48 S. fradiae cell growth and neomycin synthesis [12-14].”

This is another example, where the statement and citations are incorrect. The text has nothing in common with the references (see below).

Xing, X.H.; Wu, H.; Luo, M.F.; Wang, B.P. Effects of organic chemicals on growth of Methylosinus trichosporium OB3b. Biochem. 315 Eng. J. 2006, 31, 113-117. 316 13. Han, B.; Su, T.; Wu, H.; Gou, Z.; Xing, X.H.; Jiang, H.; Chen, Y.; Li, X.; Murrell, J.C. Paraffin oil as a "methane vector? For rapid 317 and high cell density cultivation of Methylosinus trichosporium OB3b. Appl. Microbiol. Biot. 2009. 318 14. Myung, J.; Kim, M.; Pan, M.; Criddle, C.S.; Tang, S. Low energy emulsion-based fermentation enabling accelerated methane 319 mass transfer and growth of poly(3-hydroxybutyrate)-accumulating methanotrophs. Bioresour. Technol. 2016, 207, 302-307. 320 15. Park, S.; Shah, N.N.; Taylor, R.T.; Droege, M.W. Batch cultivation of Methylosinus trichosporium OB3b: II. Production of particu- 321 late methane monooxygenase. Biotechnol. Bioeng. 1992, 40, 151-157

Line 49-50:

“In addition, some studies have  shown that ammonium ions can also affect cell growth and metabolism.”

Citation as proof is missing.

Line 49-51:

In addition, some studies have shown that ammonium ions can also affect cell growth and metabolism. It was found that nitrogen source affected cell growth and nitrate depletion caused biphasic growth pat- terns in the batch cultures of the OB3b strain (15).

What is the rational for the comparison of S. fradiae with Methylosinus trichosporium?

Park, S.; Shah, N.N.; Taylor, R.T.; Droege, M.W. Batch cultivation of Methylosinus trichosporium OB3b: II. Production of particu- 321 late methane monooxygenase. Biotechnol. Bioeng. 1992, 40, 151-157.

Line 52-53:

“…bacteria have evolved complex mechanisms…”

What kind of mechanisms?

A short description explaining it should be included. The text is very confusing, as it first refers to “complex mechanisms…” in bacteria (Ref. 16), but the next sentence guides the reader to cancer cells (Ref. 17). Consequently, the reader “must go for a journey” from S. fradiae (Streptomyces) to cancer cells. Does the comparison (without delivering the link between this mechanisms) make sense? As bacteria and eukaryotes are completely different, it is difficult to imagine that the mechanisms are similar…  

16. Commichau, F.M.; Forchhammer, K.; Stulke, J. Regulatory links between carbon and nitrogen metabolism. Curr. Opin. Microbiol. 323 2006, 9, 167-172. 324

17. Yang, M.; Vousden, K.H. Serine and one-carbon metabolism in cancer. Nat. Rev. Cancer 2016, 16, 650-662.

Line 57-60:

“The biosynthesis of neomycin B in S. fradiae is mainly regulated by the neo gene clusters, which contain two operons, one com- prising 12 genes from neoE to neoD (responsible for neomycin B biosynthesis) and the other containing the neoGH-aphA and other regulatory genes”

The biosynthesis is regulated by regulators (proteins, RNAs…), but not by genes or a whole gene cluster. In this case, there are probably regulator genes in the cluster, which encode the regulators (often specific regulators).

Line 79-82:

“For example, the comparative transcriptomic analysis revealed that the transcription factor RosR could regulate the L-glutamate metabolism and synthesis network by interacting with promoter regions of many related genes [22]. Hence, this study firstly investigated the effects of different inorganic salt ions on neomycin B synthesis by S. fradiae. It was determined that 60 mM (NH4)2SO4 most effectively promoted the biosynthesis of neomycin B and cell growth.”

There is no information about the organism in which this study was done. The reader must check the reference (see below) and is redirected to a study in a Chlorella strain.

Again, the text compares biological processes in S. fradiae with Chlorella. Chlorella is a green alga and no related to actinomycetes. As those are very different organisms, it does not fit to the manuscript that should present effects on the production of a secondary metabolite (antibiotic) (comparison of an antibiotic production in bacteria with green algae). The organisms are too different to argue that the metabolism works the same.

“It was determined that 60 mM (NH4)2SO4 most effectively promoted the biosynthesis of neomycin B and cell growth.”  

Does it mean, in this study or in general for actinomycetes? 

Zhou, L.; Cheng, D.; Wang, L.; Gao, J.; Zhao, Q.; Wei, W.; Sun, Y. Comparative transcriptomic analysis reveals phenol tolerance 335 mechanism of evolved Chlorella strain. Bioresour. Technol. 2017, 227, 266-272.

Line 81-89:

“A comparative transcriptomic analysis was  then conducted and revealed that the mechanism of (NH4)2SO4 affects neomycin B po- tency.

Based on the analysis of comparative transcriptomic and neo gene clusters overex-  pression, the neomycin B potency of 17399 U/mL at shake-flask level was achieved by overexpressing the neoE gene and supplying 60 mM (NH4)2SO4, which was 51.2% higher  than that of the control S. fradiae SF-2. The study discloses (NH4)2SO4's effect on neomycin  B synthesis through transcriptomics, accelerating neomycin B production and applica-  tions.”

Is it usual to speak about ammonium sulfate mechanism? In my opinion it is an effect that is leading to changes in the cell (e.g. sensing of ammonium sulfate/ concentration-dependent response).

It is not clear is already published results are described in the introduction or the results of this paper (own results in introduction is unusual as they should be presented in the results section).

The description is not clear. What was the result of transcriptomics? Is it the overexpression of the genes in neo cluster? What does it mean?:

the neomycin B potency of 17399 U/mL at shake-flask level was achieved by overexpressing the neoE gene and supplying 60 mM (NH4)2SO4, which was 51.2% higher than that of the control S. fradiae SF-2.”

Does it mean that the after addition of 60 mM (NH4)2SO4 the neo production was higher (51,2% increase versus control)?

“The study discloses (NH4)2SO4's effect on neomycin  B synthesis through transcriptomics, accelerating neomycin B production and applica-  tions.”

Transcriptomics cannot accelerate the production and application of an antibiotic. Transcriptomics is a method…

Line 147-150:

“…NaCl, KCl, (NH4)2SO4, and K2SO4…”

“the potency of neomycin B”

Why these salts and not others were tested? What does it mean neomycin potency? It should be defined at the beginning.

Line 163:

Figure 1 “The effects of different inorganic salt ions on the synthesis of neomycin B.”

It is impossible to understand Fig. 1.

How is it possible to measure the effect on biosynthesis? The figure shows potency against salt concentration. Probably the product neomycin was analyzed but how? In material and methods, the reader is referred to another paper, but if this should be the main results it is necessary to describe the method of detection and the procedure. It is not clear, what the figure is showing. Is it the same culture to which the salts are added? Such experiments make no sense. Each culture needs to be monitored over the whole fermentation period (grow curves plus production curves over time for each concentration). This is important, as the streptomycetes are difficult to handle and usually it is extremely difficult to obtain perfect curves for shaking flasks.

What is analyzed in Fig. 1? Is it the control strain, is it a mutant? Controls without addition of the salts are missing.

Also in the results, the sentences contain grammatic and factual errors.

Example (Line 154-156):

  “According to the results, it is not difficult  to find that 60 mM (NH4)2SO4 produces neomycin more efficiently than other test groups,  which might be attributed to three aspects.”

The salt (60 mM (NH4)2SO4) cannot produce an antibiotic (it is the strain, but the details are not described in the manuscript).

Line 169-171:

“The inorganic nitrogen (NH4)2SO4 can facilitate the con-  version of α-ketoglutarate to L-glutamate by the TCA cycle, which is then transformed into L-glutamine by transamination, thereby promoting cell growth”

For the conversion of α-ketoglutarate to L-glutamate amino acids and enzymes (transaminases) are required (not inorganic ammonium sulfate).

Line 172:

“…the specific  growth rate…”

What is specific growth rate? 

Line 200-204:

Figure 2:

2A: What is µ? How were the measurements done? What is detected? The measurement of OD for streptomycetes in complex media is not possible.   

“Firstly, it was found that 60 mM (NH4)2SO4 promoted S. fradiae SF-2 growth, causing the cells to enter the logarithmic growth phase more quickly and reach a maximum spe- cific growth rate of 0.122 h-1 at about 24 h (Figure 2A)”

The result would mean that the growth is reduced immediately after approximately 24 h. That is very surprising as a fermentation with streptomyces for antibiotic production takes at least 5 days (usually longer). The biomass is increasing in that phase.  

Figure 2:

It is impossible that the biomass is going down after short time (approx. 24 h for cultures with ammonium sulfate and approx. 40h for cultures without ammonium sulfate) and the potency of neomycin starts to increase after approx. 40 and 80 h, respectively. How is it working, if according to Fig. 2A the growth is reduced after short time?

For Fig2A, there is only one curve. Where are the other results for the replicates?   

Fig. 2B: Where are the other results for the replicates (red curve)?   

Line 198-240:

“In conclusion,  it was proved that (NH4)2SO4 could improve neomycin B potency in S. fradiae by promot- ing the utilization of carbon and nitrogen sources.”

This conclusion is based on speculations. The presented data show the growth, potency of neomycin and concentration of reducing sugars. Utilization of nitrogen sources is not included.

“The results showed that a total of 5902 genes were expressed in the medium with 60 mM (NH4)2SO4 compared with (NH4)2SO4-free culture condition of which 637 were specifically expressed …”

“(NH4)2SO4-free culture”: Is this correct. As the media are complex it is probably not possible to prepare a ammonium sulfate free medium.  

How many were expressed at the condition without ammonium sulfate? What does it mean specifically expressed? Based on the results approx. 600 genes are gone in the sample without addition of ammonium sulfate. What happened to these genes? It indicates a technical problem as these genes were not deleted, it is the same strain grown under different conditions.  

The quality of Figure 3 is not very good. It is difficult to read it.

The results indicated that 60 mM (NH4)2SO4 significantly influenced the metabolism of S. fradiae SF-2.”

This conclusion is not correct as the figure 3A only shows numbers. There is no information about the genes (genes encoding metabolic proteins).

“Based on the analysis of genes with significant changes in the transcription level us- ing the KEGG database, it was found that the genes related to the metabolic system were  mainly concentrated in the amino acid metabolism, carbohydrate metabolism, glycans  synthesis and metabolism, cofactors and vitamins metabolism, nucleic acid metabolism and energy metabolism, etc. The genes related to the genetic information processing sys- tem are mainly concentrated in translation, replication, and repair…”

It seems to affect everything.

“As a result of the analysis, it was also determined that there was a significant difference in the TCA cycle, oxidative phosphorylation, amino acid metabolism, propionate metabolism, carboxylic acid metabolism, polyketose metabolism,  the biosynthesis of vancomycin and staurosporine (Figure 3D)…”

To the best of my knowledge, S. fradiae cannot produce vancomycin. Vancomycin is mainly produced by Amycolatopsis. If there is no vancomycin cluster and production, how it is possible that there is a significant difference in this pathways?

“The transcription levels of these genes were significantly upregulated, favoring  the accumulation of precursors involved in neomycin biosynthesis

The analysis only shows the changes in the gene expression. There are no data showing the precursor accumulation.

Moreover, the transcription levels of the neo gene cluster that regulated 238 neomycin synthesis showed an upregulated trend compared to the non-added (NH4)2SO4”

The “non-added (NH4)2SO4” was not shown in Fig. 5.

In Figure 4C “Human diseases” are listed. This is strange for a bacterial strain such as S. fradiae is? This comparison seems not to be very useful. 

Figure 5B and Figure 2C are the same (the same outcome). It is not necessary to show it twice.

Is there any reason for the color gradient in Fig. 5A? The control (no addition of ammonium sulfate) is missing. What is CK?

In Figure 4 the control is also missing. It would be more important to see the difference between the two conditions (with and without ammonium sulfate).

Raw data should be included in supplementary information.

Extensive English edits are required to the manuscript.

Examples:

NaCl, KCl, (NH4)2SO4, and K2SO4 “The precursor compounds and energy generate by primary metabolism can be used to synthesize a variety of secondary metabolites with complex structures, diverse functions, and biological activities.”

This sentence is grammatically incorrect. (Probably: “generated”)

In addition, it is difficult to understand, what the message of this sentence is. Biological activity is a function. (Probably: diverse functions, including biological activities)

Line 34-35:

“Neomycin, a secondary metabolite first isolated from S. fradiae in the 1940s, is a classical aminoglycoside antibiotic made from carbohydrates by the HMP pathway”

The punctuation should be corrected.

HMP must be introduced.

Line 36-37:

“It includes three classes, A, B, and C, each with its chemical structure and biological activity.”

What does it mean? Does each class introduce a different biological activity?

Line 37-X:

“Especially for neomycin B, the main active ingredient in neomycin, it exhibits higher antimicrobial activity than other classical antibiotics (streptomycin and bacitracin, etc.) that could bind to the 16S rRNA site of the 30S ribosome and interfere with the synthesis of bacterial proteins [5-7].”

“In recent years, with the development of medical diagnostic and clinical intervention technology, the potential function of neomycin B in the medical field has become increasingly apparent.”

Strange constructions of the sentences.

Line 64-66:

“Moreover, an im- portant conserved enzyme in synthesizing 2-DOS is NeoE, a zinc-containing dehydrogen- ase that utilizes NAD(P)+ as a coenzyme [19]”

“conserved enzyme in synthesizing”

Hence, this  study firstly investigated the effects of different inorganic salt ions on neomycin B synthe- sis by S. fradiae

“this  study firstly investigated”: a study cannot investigate…

Something cannot be investigated by a bacterial strain…

Is a strange constructions…

Line 90-144:

The “Materials and Methods” section must be optimized by a scientific native speaker. Please find below an example.

Sentences such as:

“All the recombinant plasmids constructed must be transformed into Escherichia coli 98 DH5α and identified by colony PCR and sequencing at the beginning.”

The reader expects that this was done. Such steps are Good Laboratory Practice (GLP) which is a requirement for publication in peer-reviewed journals. The constructs are usually sequenced to confirm the identity of the sequence. Here it is again not clear if it was done or not. According to GLP the constructs are cloned, isolated from the cloning strain (E.coli), digested by restriction enzymes (control digestion) and sequenced to check for mutations. Only correct constructs are used for further procedures.   

“S. fradiae SF-2, E. coli DH5α and E. coli ET12567 were deposited in the laboratory.”

The Good Documentation Practices are a requirement for any scientific work (obviousness).  

“…S. fradiae SF-2 was grown with AS-1 solid medium at 35°C…”

S. fradiae and other streptomycetes are usually incubated at 28°/29°C. What is the reason for this high temperature?

Table 1 presents a long list of plasmids/strains for the expression (?) of different genes. However, there is no information what the gens are and why there were used in this study… If the reader reads the introduction and subsequently the methods section, it is difficult to understand it. 

The constructs make no sense as they are not appearing in the results. In this work only the producer (S. fradiae) was used and applied in production assays. Two sets were analyzed: with and without addition of ammonium sulfate. Wat was the purpose of these constructs?  

It is also unclear (so far /while reading the manuscript) if it is a transcriptional analysis or transcriptomics. “Omics” means the whole transcriptome was analyzed and not the expression of single genes.  

Next, the cultured cells were transferred to a fermentation medium with 60 mM (NH4)2SO4 or not…”

This would mean that the cells were transferred, and the control was not transferred to the medium…

These are only few examples. There are grammatic mistakes. Some of the sentences make no scientific sense or contain factual mistakes (e.g. Line 136: “donor cells: receptor cells=10:1” it should read recipient not “receptor”). The “Materials and Methods” section must be optimized by a scientific native speaker to correct the scientific parts and the grammar.

Author Response

Thank you very much for your suggestions, which are great and helpful to us. We have revised the manuscript thoroughly according to your advice using the “Track Changes”.

Reviewer 2 Report

Authors are asked to : 

Update that: Streptomyces fradiae is a species of Actinomycetota Phylum.

Note that Streptomyces fradiae   produces also tylosin (anti-tumoral effect) and fosfomycin

Add precision: Neomycin (aminoglyosid antibiotic) is bactericidal against both   Gram-positive and Gram-negative bacteria including S. aureus, Escherichia coli, Haemophilus influenzaeProteus spp., and Serratia spp. It is generally not effective against Pseudomonas aeruginosa

Line103: This strain is filamentous, it would have been preferable to seed spores for a better standardization of inoculates.

Line 105: It is generally accepted that the incubation temperature of this telluric bacterium is 25-28, how do you explain the temperature used in your study?

Line 148: What is  number of combinations and replicates of different concentrations (0, 20, 40, 60, 80 mmol/L) of NaCl, KCl, (NH4)2SO4, and K2SO4.  Authors followed one-factor-at-a-time (OFAT) method-like and not response surface methodology (RSM) approach!  Why?

Line 158: To control the fed-batch cultures, culture phase has to be divided into three sections with different RC/N ratios : initial, exponential, and neomycin production.

Line 207 :  Why did the cells fermented for only 48 hours with and without 60 mM (NH4)2SO4 supplementation (To investigate the mechanisms of (NH4)2SO4 effect on the neomycin B synthesis)?

Line 259:  More details should be provided to make the understanding of the figure 4 clearer (The gene expression levels .....)

Line 263:  What’is the evidence that Entner-Doudoroff (ED) pathway is nonfunctional in Streptomyces fradiae ?

 A metabolic shift  was reported (https://doi.org/10.11282/AEM.71.5.2294-2302.2005 ) when S. tenebrarius was grown on glucose and the cells passed from the growth phase to the production phase.  A change in metabolism has previously been observed with several other streptomycetes. It is apparent that the cells adjust their metabolism to the change in function: from growth to secondary metabolite production.

On other hand, NADPH is regarded as the cofactor necessary for biosynthesis of several antibiotics: β-lactams, polyketides, and glycopeptides. NADPH is produced in the PP and ED pathways but not in the EMP pathway.

Line 268: The engineered S. fradiae SF-neoE strain was fermented for 168 h .... Isn’t this time still so long (7 days)?

Author Response

(The authors gave the same response as above.)

Reviewer 3 Report

The authors have done a lot of complex and interesting work with a difficult object for molecular genetic research. However, I would like to see more understanding of the results obtained, since work has been underway in this area for many years. It seems to me that the authors have not done enough work with literature, some literary sources are almost random. A serious revision in the field of English is required. Pay attention to the Materials and Methods section.

Lines 30-31

Streptomyces fradiae belongs to the bacterial domain, such as actinomycetes and streptomycetes

Why not specify the taxonomic position more precisely? Please, clarify it with https://lpsn.dsmz.de/

Line 34

Neomycin, a secondary metabolite first isolated from S. fradiae ???

It is necessary to indicate the number of this strain in the international collection.

Line 35

“in the 1940s..”

A link should be placed here, so that it is clear who isolated the substance from this strain

Line 36

is a classical aminoglycoside antibiotic made from carbohydrates by the HMP pathway [5].

In this source there is no information about the pathway of neomycin biosynthesis

Lines 48-49

Recent studies have reported optimizing the fermentation media to promote S. fradiae cell growth and neomycin synthesis [12-14].

These links relate to studies of methanotrophic bacteria. Here the authors talk about S. fradiae.

Line 52

It was found that nitrogen source affected cell growth and nitrate depletion caused biphasic growth patterns in the batch cultures of the OB3b strain [15].

What is the relationship between Streptomyces fradiae and Methylosinus trichosporium OB3b?

Please check the use of articles.

Lines 55-57

The results of these studies indicated that nitrogen metabolism and its metabolic intermediates might affect the central metabolism of methanotrophs, thereby controlling their growth.

The fact that the composition of the medium, especially quantitative ratio of carbon and nitrogen sources, significantly affects the biosynthesis of antibiotics, has been known for a very long time. There have been a lot of such works in the past. There is no need to persistently insert works about methanotrophs here, unless of course the authors want to artificially raise the citation index of these works.

Lines 60-62

“At present, molecular biology and other technical methods are currently being studied to improve neomycin production by enhancing its biosynthesis pathway”.

Molecular biology is a science, not a method. The sentence is unsuccessful both in form and in meaning, it should be changed.

Lines 75, 77, 79

References 20-22 are not relevant to the topic of this work. There is no need to prove the applicability of comparative transcriptomics, this approach has long been used to study the regulation of secondary metabolism in actinomycetes. The authors could at least cite works on the study of streptomycetes, which use this methodology, and not algae or fungi.

Line 94

S. fradiae SF-2

I would like to see more information about the origin of this strain (from where and by whom it was isolated, how it was identified as S. fradiae etc.)

Line 105

“..75% RH...”

…needs explanation

Lines 140-144

2.5. Analytical methods

Section 2.5 is too short. I think it would be better to give a more specific title, than "analytical methods" - which can mean anything

Line 148, Figure 2

…concentrations (0, 20, 40, 60, 80 mmol/L)…

Please explain what is zero concentration? The substance was not added? Then this absence, it can be considered as a control. Please make corrections here and further, including drawings -- "making in zero concentration" is absurd

Lines 207-212

This fragment is already contained in “Materials and Methods”

Figure 3

An significant figure, but it is presented in low quality. I would like the authors to work on improving it.

Also, the authors should pay more attention to part C: the list of genes indicates those that are associated with human diseases (endocrine diseases, cancer, etc.). This is strange, since we are talking about the genome of actinobacteria.

I would like more explanations to the figure3 C, D, E

Figure 3C: How would you comment, that the genes involved in transcription are characterized by a small number of Unigens? Despite the fact that you influenced this process and the transcriptome of the strain changed under the influence of ammonium sulfate?

Lines 219-226

….The genes related to the genetic information processing system are mainly concentrated in translation, replication, and repair…

Really? Is this the first time established within the framework of this work?

Figure 4

A very interesting illustration, however, it needs a lot of explanation. Some genes have numbers highlighted in green next to them, and others in red, what does this mean? Explanations should be present in the caption to Fig.4.

Lines 284-285

In summary, uncovering the mechanism of (NH4)2SO4 effect on neomycin B synthesis in S. fradiae is beneficial for accelerating neomycin B production and applications.

In the end, what is the mechanism of action of (NH4)2SO4 on neomycin B biosynthesis?

Author Response

(The authors gave the same response as above.)
